# Dose-response relationship between diarrhea quantity and mortality in critical care patients: A retrospective cohort study

Ryohei Yamamoto[1], Hajime Yamazaki[2], Shungo Yamamoto[1], Yuna Ueta[3], Ryo Ueno[4], Yosuke Yamamoto[1]*

1 Department of Healthcare Epidemiology, School of Public Health in the Graduate School of Medicine, Kyoto University, Yoshida-honmachi, Sakyo-ku, Kyoto, Japan, 2 Section of Clinical Epidemiology, Department of Community Medicine, Graduate School of Medicine, Kyoto University, Shogoin-kawaramachi, Sakyo-ku, Kyoto, Japan, 3 Department of Nutrition Management, Kameda Medical Center, Kamogawa, Japan, 4 The Australian and New Zealand Intensive Care Research Centre, Melbourne, VIC, Australia

* yamamoto.yosuke.5n@kyoto-u.ac.jp

## Abstract

### Background

Previous studies have shown that diarrhea, defined as a dichotomized cutoff, is associated with increased mortality of patients in intensive care units (ICUs). This study aimed to examine the dose-response relationship between the quantity of diarrhea and mortality in ICU patients with newly developed diarrhea.

### Methods

We conducted this single-center retrospective cohort study. We consecutively included all adult patients with newly developed diarrhea in the ICU between January 2017 and December 2018. Newly developed diarrhea was defined according to the World Health Organization definition. The consistency of diarrhea was evaluated by the Bristol stool chart scale, and the quantity of diarrhea was assessed on the day when patients newly developed diarrhea. The primary outcome was in-hospital mortality. The risk ratio (RR) and 95% confidence interval (CI) for the association between diarrhea quantity and mortality were estimated using multivariable modified Poisson regression models.

### Results

Among the 231 participants, 68.4% were men; the median age was 72 years. The median diarrhea quantity was 401g (interquartile range [IQR] 230–645g), and in-hospital mortality was 22.9%. More diarrhea at baseline was associated with higher in-hospital mortality; the adjusted RR (95% CI) per 200-g increase was 1.10 (1.01–1.20), p = 0.029. In sensitivity analyses with near quartile categories of diarrhea quantity (<250g, 250–399g, 400–649g, ≥650g), the adjusted RRs for each respective category were 1.00 (reference), 1.02 (0.51–2.04), 1.29 (0.69–2.43), and 1.77 (0.99–3.21), p for trend = 0.033.

**Data Availability Statement:** Data cannot be shared publicly because the data contain potentially identifying or sensitive patient information.

However, data are available from the steering committee of the study (Kyoto University Graduate School and Faculty of Medicine, Ethics Committee) for researchers who meet the criteria for access to confidential data (contact via: 060kensui@mail2. adm.kyoto-u.ac.jp).

**Funding:** The authors received no specific funding for this work.

**Competing interests:** The authors have declared that no competing interests exist.

## Conclusions

A greater quantity of diarrhea was an independent risk factor for in-hospital mortality. The diarrhea quantity may be an indicator of disease severity in ICU patients.

## Introduction

Diarrhea is a common gastrointestinal symptom in the intensive care unit (ICU), with an incidence of 35%–70% [1]. In ICU patients, enteral nutrition (composition, osmolarity, speed, intermittent or continuous, and fiber), drugs (e.g., antibiotics, laxatives), infectious diseases (e.g., *Clostridium difficile* infection [CDI]), and comorbidity (e.g., anemia, cirrhosis) can cause diarrhea [2]. The effects of diarrhea include increased risk of contamination of devices and wounds, dehydration, electrolyte abnormalities, and malabsorption [3–5].

Several studies have shown an association between diarrhea and mortality [6–14], and this association remained even in ICU patients without CDI [8]. Taito et al. conducted a systematic review and demonstrated that diarrhea was associated with the length of hospital stay and ICU mortality [14]; however, all previous studies defined diarrhea based on dichotomized criteria with respect to consistency and frequency. The European Society of Intensive Care Medicine (ESICM) has adopted dichotomized criteria for the quantity of diarrhea as a component of the definition of diarrhea in the ICU [3]. However, healthcare providers need to make decisions based on continuous conditions rather than dichotomized conditions in practice [15, 16]. For example, more diarrhea may cause worse electrolyte imbalance, nutritional deficit, and hemodynamic instability owing to water loss [17, 18], which leads to changes in clinical management. Moreover, more diarrhea may result in more deaths. To clarify this, it is necessary to quantify the relationship between the quantity of diarrhea and death.

This retrospective cohort study aimed to examine the dose-response relationship between diarrhea in ICU and mortality. We investigate the association between the quantity of diarrhea and mortality in ICU patients with newly developed diarrhea.

## Materials and methods

### Study design and setting

We conducted this single-center retrospective cohort study at Kameda Medical Center ICU. This study was reviewed and approved by the institutional review board of Kyoto University (R2253) and Kameda medical center (19–145). These committees waived the requirement of informed consent from all participants enrolled in this study because of the retrospective study design. This study was conducted according to the Strengthening the Reporting of Observational Studies in Epidemiology (STROBE) guidelines [19]. A preprint has previously been published [20].

### Study population

From January 2017 to December 2018, we consecutively included all patients aged ≥18 years with newly developed diarrhea in the ICU. We defined newly developed diarrhea in the ICU as three or more loose or liquid stools per day according to the World Health Organization (WHO) definition [21]. To include patients with newly developed diarrhea in the ICU, the following patients were excluded on the day of ICU admission: patients with a stoma, chronic diarrhea (e.g., inflammatory bowel syndrome, short bowel syndrome), post-gastrointestinal surgery, gastrointestinal bleeding, or bacterial and viral enteritis (including *Clostridium*

*difficile* enteritis and cytomegalovirus enteritis) or those who already had diarrhea on the day of ICU admission. In addition, patients readmitted to the ICU and those who died on the day of admission were excluded.

## Data collection

We collected data such as age, sex, admission category (medical or surgery), sepsis defined by sepsis-3 [22], ICU readmission, Charlson Comorbidity Index (CCI) [23], and treatment limitation (limitations in providing ICU-specific life-sustaining therapies such as mechanical ventilation, cardiopulmonary resuscitation) from electronic health record reviews. Other data such as Acute Physiology and Chronic Health Evaluation (APACHE) II score [24], Simplified Acute Physiology Score (SAPS) II [25], Sequential Organ Failure Assessment (SOFA) score [26], potential causes of diarrhea (proton pump inhibitor, enteral nutrition, antibiotics, laxative drugs), testing for CDI (glutamate dehydrogenase test, CD toxin, or stool culture), and biopsy-diagnosed cytomegalovirus enteritis were also collected. We refer to CCI with age score as "CCI" and define CCI without age score as "CCI without age score."

## Measurement of diarrhea

We defined diarrhea by the WHO definition (three or more loose or liquid stools per day). Stool data of ICU patients were collected from electronic health records. Nurses routinely checked the presence or absence of stools every 2–4 h during ICU stay. In all ICU patients, the consistency and quantity of all stool samples were assessed by a nurse. The Bristol Stool Chart Scale (BSCS) was used to evaluate the consistency of each stool sample [27]. The BSCS is a 7-point scale in which stools are scored according to cohesion and surface cracking as follows: 1. separate hard lumps like nuts; 2. sausage shaped but lumpy; 3. like a sausage or snake but with cracks on its surface; 4. like a sausage or snake and smooth and soft; 5. soft blobs with a clear-cut edge; 6. fluffy pieces with ragged edges and mushy; and 7. watery with no solid pieces. This scale has been evaluated for its concordance and is a widely used scale [28–34]. The quantity of stool was measured using a weight scale and recorded in the electronic medical records. Since most ICU patients had urinary catheters, contamination of the stool by urine was minimized. The main exposure was the quantity of diarrhea per day on the day of the diarrhea diagnosis. The daily quantity of diarrhea was calculated from calendar days (total quantity from 0:00 to 24:00). These data were collected from electronic health records.

## Outcome measurement

The primary outcome was in-hospital mortality. Secondary outcomes included ICU, 28-day, and 90-day mortality; ICU-free days at the 28-day [35]; and hospital-free days at the 90-day [36]. In these free days, we used the event-free survival day (the number of event-free days was considered zero for patients who died in the time frame) to measure these outcomes.

## Statistical analyses

Patient characteristics are described as median and interquartile range (IQR). Modified Poisson regression models were used to estimate risk ratios (RRs) and 95% confidence intervals (CIs) for the association between the quantity of diarrhea (per 200-g increase) and in-hospital mortality [37, 38]. The reason for using the unit of 200 g is that the ESICM uses a 200–250 g cutoff for diarrhea quantity [3]. The multivariable analysis was adjusted for CCI, SOFA score, and serum albumin levels. These covariates were selected a priori based on clinical plausibility and previous studies [2, 8, 13]. We performed multiple imputations for missing values using

multiple imputations by chained equation (MICE) with 50 iterations that generated 100 data-sets with imputed missing values [39, 40].

To perform sensitivity analyses, we tested several modified Poisson regression models to assess the robustness of the primary analysis. First, we adjusted for the following covariates: model 1 for age, sex, CCI without age score, SOFA score, and serum albumin; model 2 for CCI, APACHE II score, and serum albumin; model 3 for CCI, SAPS II score, and serum albumin; model 4 for CCI, SOFA score, serum albumin, and enteral nutrition; and model 5 for CCI, SOFA score, serum albumin, enteral nutrition, and laxatives. Second, we conducted a complete case analysis. Third, because CDI and cytomegalovirus enteritis affects mortality, we performed a further analysis excluding patients diagnosed with them after ICU admission. Fourth, to evaluate the influence of urine and stool contamination, we performed an additional analysis limited to only patients without any risk of urine contamination (anuric patients or patients with any forms of urinary cathe-ter and/or nephrostomy). Finally, instead of using continuous data (the actual quantity of diarrhea), we used categorical data (near quartiles of the quantity of diarrhea) for the primary model. We calculated P for trends across the median values of each near quantile category [41].

We applied the same analyses as those for the primary outcome to assess the association between the quantity of diarrhea (per 200-g increase) and the following secondary outcomes: ICU mortality, 28-day mortality, and 90-day mortality.

We reported 95% CIs as an informal measure of uncertainty and avoided using terms such as statistical significance according to the recommendations of the American Statistical Associ-ation [42]. The analyses were performed using R software, version 4.0.3 (The R Foundation for Statistical Computing, Vienna, Austria; https://www.R-project.org/).

## Results

### Patient characteristics

During the study period, 1579 patients were admitted to the ICU, and 334 adult patients with newly developed diarrhea were included in this study (S1 Table). Among those patients, 103 were excluded. Finally, 231 patients were included in the analysis (shown in Fig 1).

The median age of patients was 72 (IQR [64, 80]) years, 158/231 (68.4%) patients were men, median CCI was two (IQR [1, 3]), median APACHE II score was 21 (IQR [14, 28]), and median SOFA score was 9 (IQR [6, 12]). Patients admitted for nonoperative reasons were the most prevalent (162/231, 70.1%). Sepsis was diagnosed in 121 patients (52.4%). Antimicrobials and laxative drugs as possible causes of diarrhea were administered to 214/231 (92.6%) and 119/231 (51.5%) patients, respectively. Overall, 2/231 (0.9%) patients were diagnosed with CDI in the ICU, and two (0.9%) patients were diagnosed with CMV by colonoscopic biopsy in the ICU. None of the patients used probiotics or synbiotics. Some forms of urinary catheter and/or nephrostomy catheters were inserted in 218/231 (94.4%) patients (216 urinary catheters and 2 nephrostomy catheters), and 5/231 (2.2%) patients were anuric. The median number of days from ICU admission to newly developed diarrhea was 3 (IQR [2, 6]), and the median quantity of diarrhea was 401 (IQR [230.5, 645]) g. The median consecutive days of diarrhea was 1 day (IQR [1, 2]), and the median total number of days of diarrhea in the ICU was 2 days (IQR [1, 4]).

Other patient characteristics on ICU admission are summarized in Table 1. Three patients had missing values for the severity score because arterial blood gas was not measured. There were no missing measurements for other variables, including the quantity of diarrhea.

### Association between the quantity of diarrhea and outcomes

Table 2 presents mortality, length of stay, and free day survival. Two and 16 patients were lost to 28-day and 90-day follow-ups, respectively. In the unadjusted analysis, the quantity of

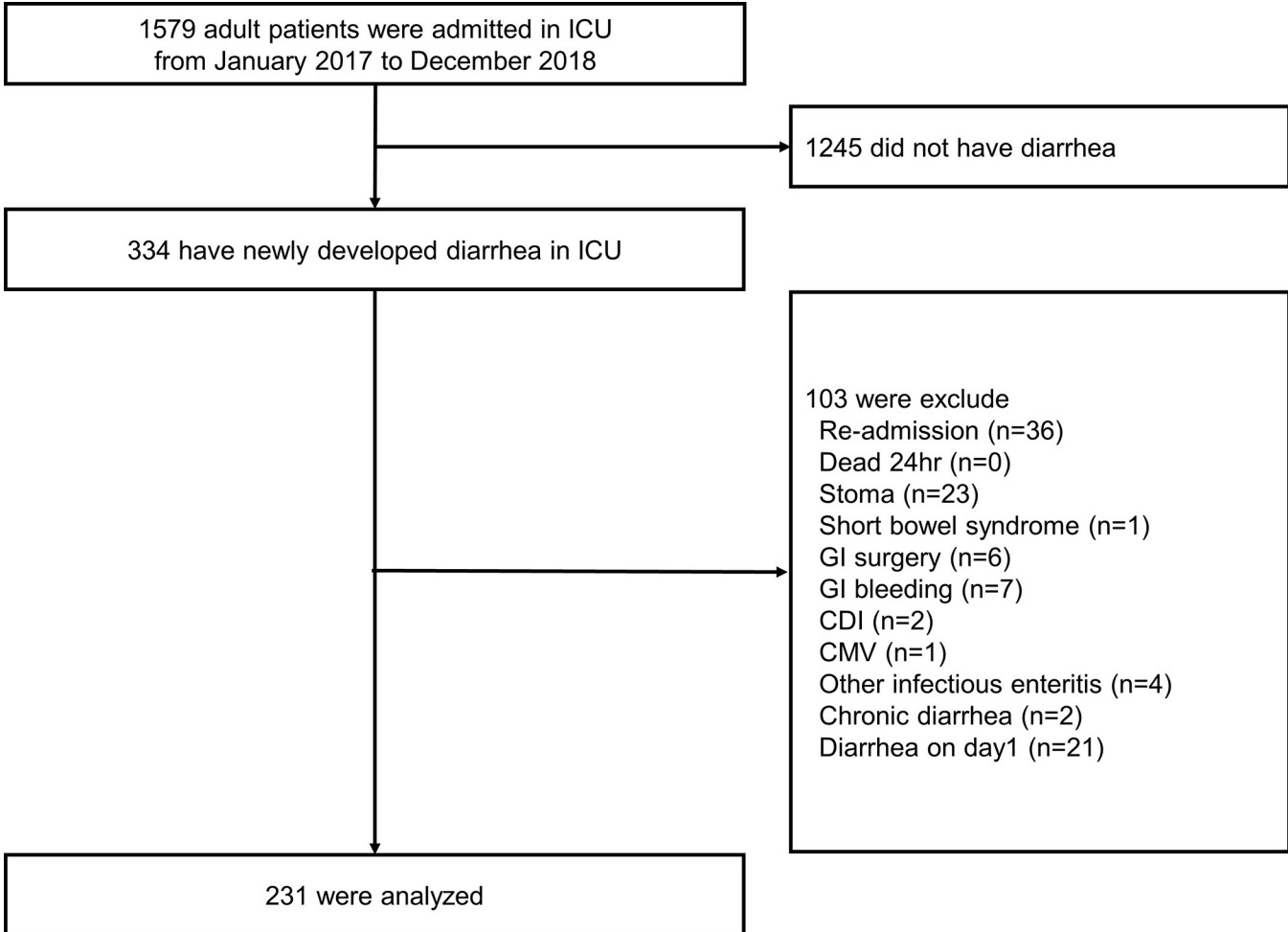

**Fig 1. Flow diagram of the sample selection.** In our ICU, nurses routinely assess the consistency and quantity of all stools; therefore, no patients were excluded due to missing stool information. GI: Gastrointestinal, CDI: *Clostridioides difficile* infection, CMV: cytomegalovirus infection.

diarrhea was associated with increased in-hospital mortality (unadjusted RR per 200 g increased: 1.10 [95% CI 1.01–1.19], p = 0.03). After adjusting for CCI, SOFA score, and serum albumin level, this association remained (adjusted RR per 200-g increase: 1.10 [95% CI 1.01–1.20], p = 0.03) (Table 3).

### Sensitivity analyses for the primary analysis

The association between the quantity of diarrhea and in-hospital mortality remained similar in various multivariable analysis models and other sensitivity analyses (Table 3). We also performed a sensitivity analysis using the categories of the quantity of diarrhea. With no established criteria to distinguish the quantity of diarrhea, we used near-quantile-defined categories of the quantity of diarrhea. The quartiles of diarrhea were 230 g in the 25th percentile, 401 g in the 50th percentile, and 645 g in the 75th percentile. Therefore, the patients were divided into the following categories: mild (<250 g), moderate (250–399 g), severe (400–649 g), and very severe (≥650 g). In-hospital mortality was 19.7% (12/61) for mild, 19.2% (10/52) for moderate, 21.3% (13/61) for severe, and 31.6% (18/57) for very severe. Multivariable-modified Poisson regression analysis using these categories, with the same adjustments as in the primary model,

**Table 1. Baseline characteristics of the study patients.**

| | Total n = 231 |
|---|---|
| Sex, males, n (%) | 158 (68.4) |
| Age, median [IQR] | 72 [64, 80] |
| Admission source, n (%) | |
| Hospital ward | 56 (24.2) |
| Emergency department | 106 (45.9) |
| Elective surgery | 39 (16.9) |
| Emergency surgery | 30 (13.0) |
| Admission category*, n (%) | |
| Post cardiovascular surgery | 50 (21.6) |
| Sepsis | 121 (52.4) |
| Pneumonia | 49 (40.4) |
| Urinary tract | 17 (14.0) |
| Abdominal | 7 (5.8) |
| Skin/Soft tissue | 7 (5.8) |
| Other | 41 (34) |
| Respiratory | 11 (4.8) |
| Neurological | 7 (3.0) |
| Metabolic | 7 (3.0) |
| Trauma | 6 (2.6) |
| Cardiovascular | 4 (1.7) |
| Hematologic | 4 (1.7) |
| Other | 21 (9.1) |
| Charlson comorbidity index, median [IQR] | 2 [1, 3] |
| Serum albumin, median [IQR], g/dL | 2.8 [2.2, 3.2] |
| SOFA score[†], median [IQR] | 9 [6, 12] |
| APACHE II score[†], median [IQR] | 21 [14, 28] |
| SAPS II score[†], median [IQR] | 48 [37, 60] |
| ARDS, n (%) | 42 (18.2) |
| Acute kidney injury, n (%) | 105 (41.1) |
| Renal replacement therapy, n (%) | 58 (25.1) |
| Mechanical ventilation, n (%) | 147 (63.6) |
| Noradrenaline, n (%) | 139 (60.2) |
| Proton pump inhibitor, n (%) | 196 (84.8) |
| Laxative drug, n (%) | 119 (51.5) |
| Antibiotics, n (%) | 214 (92.6) |
| Antiviral, n (%) | 18 (7.8) |
| Chemotherapy, n (%) | 8 (3.5) |
| Enteral nutrition, n (%) | 157 (68.0) |
| *Clostridium difficile* infection[‡], n (%) | 2 (0.9) |
| Cytomegalovirus enteritis, n (%) | 2 (0.9) |
| Quantity of diarrhea, median [IQR], g | 401 [230, 645] |
| Onset of diarrhea[§], median [IQR], day | 3 [2, 6] |
| Consecutive days of diarrhea, median [IQR], day | 1 [1, 2] |

(*Continued*)

**Table 1.** (Continued)

|  | Total n = 231 |
|---|---|
| Total number of days of diarrhea, median [IQR], day | 2 [1, 4] |

*APACHE category [43].

†Three missing data

‡Defined by glutamate dehydrogenase positivity and *Clostridium difficile* toxin positivity.

§Number of days from ICU admission to the onset of diarrhea

IQR: Interquartile range, SOFA: Sequential organ dysfunction assessment, ARDS: Acute respiratory distress syndrome, APACHE II: Acute Physiology and Chronic Health Disease Classification System, SAPS II: Simplified acute physiology score.

showed a trend toward increased in-hospital mortality as the quantity of diarrhea increased (P for trend = 0.033) (Table 3).

## Secondary analyses

For secondary analyses, a similar association was observed between the quantity of diarrhea and ICU 28-day and 90-day mortalities (Table 4). Multivariable analysis showed a similar trend of higher mortality with higher quantities of diarrhea.

## Discussion

In this retrospective study, we investigated the association between the quantity of diarrhea and in-hospital mortality in 231 patients with newly developed diarrhea in the ICU. Multivariable analysis revealed that diarrhea quantity was an independent predictor of in-hospital mortality. This association was consistent across several sensitivity analyses. Similarly, the greater the quantity of diarrhea, the higher the ICU 28-day and 90-day mortalities. To the best of our knowledge, this is the first study to show an association between the quantity of diarrhea and mortality.

In diagnosing diarrhea, the quantity, as well as the frequency and consistency, can help us predict outcomes. In this study, we showed that mortality increased with increasing diarrhea quantity, according to the adjusted RR in patients with newly developed diarrhea in the ICU.

**Table 2.** Mortality, length of stay, and free day survival of the study patients.

|  | Total n = 231 |
|---|---|
| Mortality, n/total n (%) |  |
| Hospital mortality | 53/231 (22.9) |
| ICU mortality | 21/231 (9.1) |
| 28-day mortality† | 35/229 (15.3) |
| 90-day mortality‡ | 52/215 (24.2) |
| Length of stay (LOS) and free day survival, median [IQR], day |  |
| ICU LOS | 7.0 [4.0, 12.4] |
| ICU-free day survival at 28* | 20.0 [14.0, 23.0] |
| Hospital LOS | 35.0 [18.4, 58.0] |
| Hospital-free day survival at 90† | 45.0 [0, 66.5] |

†Two patients lost to follow-up, ‡16 patients lost to follow-up. There were no missing measurements in other outcomes. IQR: Interquartile range, LOS: Length of stay

**Table 3. Association between the quantity of diarrhea and in-hospital mortality.**

| | Unadjusted | | Adjusted | |
|---|---|---|---|---|
| | RR [95% CI] | p-value | RR [95% CI] | p-value |
| **Primary analysis (per 200-g diarrhea increase)** | 1.10 [1.01, 1.19] | 0.031 | 1.10 [1.01, 1.20] | 0.029 |
| **Sensitivity analyses (per 200-g diarrhea increase)** | | | | |
| Model 1 | | | 1.09 [0.98, 1.20] | 0.080 |
| Model 2 | | | 1.10 [1.01, 1.20] | 0.028 |
| Model 3 | | | 1.11 [1.02, 1.22] | 0.018 |
| Model 4 | | | 1.10 [1.00, 1.20] | 0.041 |
| Model 5 | | | 1.09 [1.00, 1.19] | 0.048 |
| Complete case analysis | 1.10 [1.05, 1.15] | 0.001> | 1.10 [1.04, 1.17] | 0.002 |
| Exclude CDI or CMV diagnosed in ICU | 1.10 [1.03, 1.17] | 0.006 | 1.14 [1.04, 1.24] | 0.004 |
| Patients without any risk of urine contamination | 1.10 [1.01, 1.19] | 0.029 | 1.10 [1.01, 1.21] | 0.030 |
| Quantile-defined categories | | | | |
| Mild (<250 g) | 1.00 (reference) | | 1.00 (reference) | |
| Moderate (250–399 g) | 0.97 [0.38, 2.49] | 0.953 | 1.02 [0.51, 2.04] | 0.963 |
| Severe (400–649 g) | 1.11 [0.46, 2.68] | 0.823 | 1.29 [0.69, 2.43] | 0.421 |
| Very severe (≥650 g) | 1.61 [0.85, 3.04] | 0.145 | 1.77 [0.99, 3.21] | 0.056 |
| P for trend | | 0.09 | | 0.033 |

Primary model: CCI, SOFA score, and serum albumin, Model 1: age, sex, CCI without age score, SOFA score, and serum albumin. Model 2: CCI, APACHE II score, and serum albumin. Model 3: CCI, SAPS II score, and serum albumin. Model 4: CCI, SOFA score, serum albumin, and enteral nutrition, Model 5: CCI, SOFA score, serum albumin, enteral nutrition, and laxatives.

RR: Risk ratio, CI: Confidence interval, CCI: Charlson comorbidity index, SOFA: Sequential Organ Failure Assessment, CDI: *Clostridium difficile* infection, CMV: Cytomegalovirus enteritis

Previous studies have reported an association between the presence of diarrhea and mortality; however, no studies have examined whether mortality increases with a greater quantity of diarrhea [14, 44]. A systematic review of 12 studies, most of which used the definition of diarrhea as three or more loose or liquid stools, showed an association between diarrhea and mortality, but with high heterogeneity (RR: 1.43; 95% CI: 1.03–1.98; $I^2$ = 86.7%; n = 11,866) [14]. A recent prospective study evaluating the association between the presence or absence of diarrhea, as defined by the WHO definition of at least 3 liquid bowel movements per day, found that mortality was not associated with the presence of diarrhea [1]. One possible reason for the inconsistent results of these studies may be that diarrhea is defined by consistency and frequency, without taking quantity into account, which might lead to classifying small quantities of clinically unimportant bowel movements as diarrhea. More diarrhea leads to worse electrolyte

**Table 4. Association between the 200-g increase in the quantity of diarrhea and secondary outcomes.**

| | Unadjusted | | Adjusted | |
|---|---|---|---|---|
| | RR [95% CI] | p-value | RR [95% CI] | p-value |
| ICU mortality | 1.17 [1.07, 1.29] | 0.001 | 1.20 [1.07, 1.35] | 0.002 |
| 28-day mortality | 1.11 [1.01, 1.23] | 0.028 | 1.11 [0.99, 1.23] | 0.053 |
| 90-day mortality | 1.10 [1.01, 1.19] | 0.028 | 1.11 [1.01, 1.21] | 0.025 |

All analyses were adjusted for CCI, SOFA score, and serum albumin level.

RR: Risk ratio, CI: Confidence interval, CCI: Charlson comorbidity index, SOFA: Sequential Organ Failure Assessment

imbalance, nutritional deficit, and hemodynamic instability owing to water loss [17, 18], so the quantity of diarrhea should be evaluated for ICU patients.

The reason for the higher mortality rate among patients with a greater quantity of diarrhea remains unclear. Patients with CDI or cytomegalovirus enteritis, which are known to cause diarrhea, have been reported to have higher mortality, but they were excluded from our study. Indeed, diarrhea can cause dehydration, electrolyte abnormalities, metabolic acidosis, malnutrition, device contamination, and wound contamination [45]. However, since dehydration and electrolyte abnormalities are carefully corrected in the ICU, it is questionable to assume that diarrhea directly contributes to mortality.

Possible explanations for the relationship between diarrhea and mortality are as follows. First, diarrhea can be a sign of gastrointestinal organ failure that is associated with a high risk of mortality [4, 14, 46]. Patients with diarrhea have higher severity scores than those without diarrhea [2, 6, 8–10, 14]. In our study, most patients received treatments that could cause diarrhea, such as enteral nutrition and antimicrobials. These interventions are part of the treatment regimen for critically ill patients. In addition, approximately 60% of patients were on ventilation and used vasopressors, which means that patients with diarrhea have a higher severity of illness. In our analysis, we adjusted for the SOFA score, an organ disorder score that does not include gastrointestinal function, and showed that diarrhea is a risk factor for mortality independent of other organ disorders. The quantity of diarrhea may be a candidate when adjusting for organ dysfunction. Second, diarrhea can be a sign of a disorder of the gut microbiota, which is called dysbiosis. This dysbiosis is believed to increase vulnerability to nosocomial infections, sepsis, organ failure, and mortality [47, 48]. The development of diarrhea might be associated with dysbiosis in the gut microbiota of ICU patients [49]. However, our data and analyses are not sufficiently conclusive to prove them. Further research is needed to test these hypotheses.

This study had several limitations. First, the measurement of diarrhea was not completely accurate. If diarrhea spills out of the diaper, it may not be measured. In this case, this may have led to an underestimation of the quantity of diarrhea. However, we believe that this measurement of the quantity of diarrhea reflects real clinical practice. Second, the inter-rater reliability of BSCS was not confirmed in our study. The reliability of BSCS has been studied and widely used [27–32, 50], and our nurses were trained to measure BSCS in clinical practice, which should have minimized the inter-rater variability. Third, eight patients were voiding freely without the use of urine catheters. Measuring the weight of the diaper could be the sum of stool and urine in those patients; however, our sensitivity analysis showed that the increase in the risk of mortality remained after limiting the analysis to only patients without risk of urine contamination. Fourth, mechanistic data on developing diarrhea in ICUs were lacking. To elucidate mechanisms, studies are needed to assess the relationship between changes in the microbiome over time during ICU stays with newly developed diarrhea, and to evaluate whether interventions to correct the microbiome improve diarrhea. Finally, this was a retrospective single-center study. Prospective multicenter studies that consider diarrhea quantity are needed to improve the measurement validity of diarrhea and the generalizability of these findings.

## Conclusion

The greater quantity of diarrhea was associated with higher mortality in ICU patients with newly developed diarrhea. The quantity of diarrhea may be considered an indicator of disease severity in ICU patients. Further research is needed to determine if there is a causal relationship between the quantity of diarrhea and death.

## Supporting information

**S1 Table. The mortality and length of stay of the entire cohort.**
(DOCX)

## Acknowledgments

The authors thank the staff of the ICU of the Kameda Medical Center. We thank the Japanese Society of Education for Physicians and Trainees in the Intensive Care Clinical Trial Group (JSEPTIC-CTG) for their suggestions and comments on the earlier concept of this study.

## Author Contributions

**Conceptualization:** Ryohei Yamamoto, Hajime Yamazaki, Shungo Yamamoto, Yuna Ueta, Ryo Ueno, Yosuke Yamamoto.

**Data curation:** Ryohei Yamamoto, Yuna Ueta.

**Formal analysis:** Ryohei Yamamoto, Hajime Yamazaki, Ryo Ueno.

**Investigation:** Ryohei Yamamoto, Yuna Ueta.

**Methodology:** Ryohei Yamamoto, Hajime Yamazaki, Shungo Yamamoto, Ryo Ueno, Yosuke Yamamoto.

**Project administration:** Ryohei Yamamoto, Hajime Yamazaki, Yuna Ueta, Yosuke Yamamoto.

**Supervision:** Hajime Yamazaki, Shungo Yamamoto, Yosuke Yamamoto.

**Validation:** Ryohei Yamamoto, Hajime Yamazaki, Shungo Yamamoto, Yuna Ueta, Ryo Ueno, Yosuke Yamamoto.

**Visualization:** Ryohei Yamamoto, Ryo Ueno.

**Writing – original draft:** Ryohei Yamamoto, Hajime Yamazaki, Yosuke Yamamoto.

**Writing – review & editing:** Ryohei Yamamoto, Hajime Yamazaki, Shungo Yamamoto, Yuna Ueta, Ryo Ueno, Yosuke Yamamoto.

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
