## [Decision Letter · Decision Letter 0]

7 Sep 2022

PONE-D-22-18062Dose-response relationship between diarrhea quantity and mortality in critical care patients: A retrospective cohort studyPLOS ONE

Dear Dr. Yamamoto,

Thank you for submitting your manuscript to PLOS ONE. After careful consideration, we feel that it has merit but does not fully meet PLOS ONE’s publication criteria as it currently stands. Therefore, we invite you to submit a revised version of the manuscript that addresses the points raised during the review process.

We look forward to receiving your revised manuscript.

Kind regards,

Chinh Quoc Luong, MD., PhD.

Academic Editor

PLOS ONE

Journal Requirements:

Additional Editor Comments:

The Authors' manuscript has been peer-reviewed by three experts in the field. I have received three completed reviews; the Reviewers' comments are available below. The Authors used a retrospective cohort to examined the association between diarrhea quantity and mortality. The topic studied by the Authors is clinically relevant and of interest to the readers of PLOS ONE. However, the Reviewers have raised several concerns which should be addressed. Based on the Reviewers' comments, I invite the Authors to submit a revision of their paper that addresses the points raised during the review process.

Reviewers' comments:

Reviewer's Responses to Questions

**Comments to the Author**

1. Is the manuscript technically sound, and do the data support the conclusions?

Reviewer #1: No

Reviewer #2: Yes

Reviewer #3: Partly

2. Has the statistical analysis been performed appropriately and rigorously? 

Reviewer #1: Yes

Reviewer #2: Yes

Reviewer #3: Yes

3. Have the authors made all data underlying the findings in their manuscript fully available?

Reviewer #1: No

Reviewer #2: No

Reviewer #3: Yes

4. Is the manuscript presented in an intelligible fashion and written in standard English?

Reviewer #1: Yes

Reviewer #2: Yes

Reviewer #3: Yes

5. Review Comments to the Author

Reviewer #1: Thank you for giving me the opportunity to review this paper.

Major points:

1) Since the use of laxatives and enteral feedings are considered to be directly related to the amount of diarrhea, should they be included as factors for adjustment?

2) The diagnosis is unclear when the patient enters the ICU. Pneumonia? Trauma? Poison? Postoperative for malignant tumor? etc.

3) Figure1: Although it is a retrospective study, the amount of diarrhea was measured in all cases, and no cases were excluded due to missing data. Do you always measure the amount of diarrhea in all patients with diarrhea? If so, this needs to be clearly stated.

4) Table 2: Table2 is not the primary and secondary endpoint, but simply presents the mortality rate, etc., of the patients in the analysis.

Minor points:

1) What did the physician assess as the cause of diarrhea in these patients?

2) Were there no probiotics or synbiotics done?

3) Were any drugs such as Daiken Chutou used to regulate bowel movements?

Reviewer #2: The manuscript is quite interesting as it offers a quantitative approach to diarrhea in the ICU. However, I wonder about a few points:

1. The technique of determining the amount of stool based on a retrospective electronic medical record may not be accurate. In fact, not all ICU cases are placed on a urinary catheter, and weighing the diaper will mix feces and urine.

2. Diarrhea is the result of some medical conditions, some drugs used and even the diet. Can the authors further analyze the correlation between the amount of diarrhea and these factors?

3. Quantification of diarrhea should note the number of days of diarrhea and duration of diarrhea. In the article, only the amount of stools per day was mentioned, but not the whole process of diarrhea.

Reviewer #3: Thank you for the opportunity to review this interesting manuscript. The authors used a retrospective cohort to examined the association between diarrhea quantity and mortality. Below are a few comments intended to enhance the manuscript.

1.Incidence of diarrhea- the authors report the incidence of diarrhea in the ICU 10-78%. The studies quoted are older studies. The most recent Prospective Cohort study (Dionne et al, ICM 2022) showed the incidence of diarrhea in the ICU to be between 35-70%. Would suggest updating this.

2. The authors state that CDI is a common cause of diarrhea in the ICU. However, both prospective and retrospective cohort studies show a rate of 2.1%. The the present cohort it was <1%. Suggest rephrasing.

3. The most recent study (PMID: 35411491) did not show an association between mortality and diarrhea. Many of the studies that demonstrated this relationship where retrospective in nature. A point that should be highlighted in the current manuscript.

4. The authors describe different definitions of diarrhea throughout the manuscript-- Bristol >6, WHO definition of >3 liquid BM per day, then quote the ESICM of >200g. Suggest clarifying this and harmonizing it in the manuscript

5. The authors highlight the limitations of the study, however, suggest expanding the discussion to include areas for further research to help answer the remaining questions in this area (ex. mechanism for diarrhea in the ICU) and how this research may be done.

6. The Mortality for diarrhea in this study was 20%. It would be helpful to the reader to compare that to typical ICU stay or compare it to the patients in the study who did not develop diarrhea.

I congratulate the authors on their work and wish them well on their future research endeavours.

6. PLOS authors have the option to publish the peer review history of their article (what does this mean?). If published, this will include your full peer review and any attached files.

Reviewer #1: No

Reviewer #2: No

Reviewer #3: No

---

## [Author Response · Author response to Decision Letter 0]

18 Nov 2022

October 8, 2022

Chinh Quoc Luong, MD., PhD.

Academic Editor

PLOS ONE

Title: "Dose-response relationship between diarrhea quantity and mortality in critical care patients: A retrospective cohort study"

Ref: Submission ID PONE-D-22-18062

Dear editors and reviewers

We truly appreciate the constructive comments and insightful suggestions from the editors and reviewers which have helped improve the quality of our manuscript. We are thankful for the opportunity to resubmit our manuscript entitled "Dose-response relationship between diarrhea quantity and mortality in critical care patients: A retrospective cohort study" for publication in PLOS ONE. We agree with your suggestions and we have revised our paper accordingly.

In accordance with your instructions, we have prepared the following files:

2) Revised manuscript with track change to show the edits

3) Revised clean copy of our manuscript, figures, and supplementary materials

We hope that we have satisfactorily addressed all the comments raised and that our paper is now acceptable for publication. We would like to again thank the editors and reviewers for their time and helpful comments and look forward to hearing from you.

Most sincerely,

Yosuke Yamamoto, MD, PhD

Department of Healthcare Epidemiology, School of Public Health in the Graduate School of Medicine, Kyoto University

Yoshida-Konoe-cho, Sakyo-ku, Kyoto 606-8501, Japan

Tel: +81-075-753-9467

Fax: +81- 075-753-4644

E-mail: yamamoto.yosuke.5n@kyoto-u.ac.jp

POINT-BY-POINT RESPONSES TO EDITORS’ AND REVIEWERS’ COMMENTS:

Reviewer 1

Reviewer comment

Thank you for giving me the opportunity to review this paper.

Major points:

1) Since the use of laxatives and enteral feedings are considered to be directly related to the amount of diarrhea, should they be included as factors for adjustment?

Response

Thank you for your interest in our work and your helpful comments. We already adjusted for enteral feeding as a covariate in our sensitivity analysis (Table 3). As suggested, we have added a sensitivity analysis adjusting for CCI, SOFA score, serum albumin, enteral feeding, and laxatives as Model 5. 

The estimate from Model 5 was similar to that from the primary analysis (RR 1.09, 95% CI 1.00-1.19, p= 0.048).

We have revised the methods (lines 147-152) and added the result in Table 3. 

Methods (lines 147-152)

“First, we adjusted for the following covariates: model 1 for age, sex, CCI without age score, SOFA score, and serum albumin; model 2 for CCI, APACHE II score, and serum albumin; model 3 for CCI, SAPS II score, and serum albumin; model 4 for CCI, SOFA score, serum albumin, and enteral nutrition; and model 5 for CCI, SOFA score, serum albumin, enteral nutrition, and laxatives.”

Reviewer comment

2) The diagnosis is unclear when the patient enters the ICU. Pneumonia? Trauma? Poison? Postoperative for malignant tumor? etc.

Response

Thank you for this comment. The diagnosis on ICU admission were Pneumonia (n=49), Trauma (n=6), Poison (n=3), Postoperative for malignant tumor (n=0).

We added the diagnosis category on ICU admission to Table 1 based on the APACHE admission category, which is well-known in ICU research (Chest. 1991;100(6):1619-36). 

Reviewer comment

3) Figure1: Although it is a retrospective study, the amount of diarrhea was measured in all cases, and no cases were excluded due to missing data. Do you always measure the amount of diarrhea in all patients with diarrhea? If so, this needs to be clearly stated.

Response

We appreciate the reviewer’s suggestions. As mentioned in the Methods (lines 114-117) and Results (lines 186-187) sections, we always measure the quantity of all stools, regardless of diarrhea, for all patients in routine practice. 

We have revised it more clearly in Methods (lines 114-117), and the legend of Fig. 1. (Lines 178-179) as below; 

Methods (lines 114-117)

“Nurses routinely checked the presence or absence of stools every 2–4 h during ICU stay. In all ICU patients, the consistency and quantity of all stool samples were assessed by a nurse.”

Fig. 1. Flow diagram of the sample selection (line178-179)

“In our ICU, nurses routinely assess the consistency and quantity of all stools; therefore, no patients were excluded due to missing stool information.”

Reviewer comment

4) Table 2: Table2 is not the primary and secondary endpoint, but simply presents the mortality rate, etc., of the patients in the analysis.

Response

As suggested, we have changed the Results section (line 205) and the Table 2 title as below, 

Results line 205

“Table 2 presents mortality, length of stay, and free day survival.”

Table 2 title

“Table 2. The mortality, length of stay, and free day survival of the study patients.”

Reviewer comment

Minor points:

1) What did the physician assess as the cause of diarrhea in these patients?

Response

Thank you for this comment. The causes of diarrhea are varied and it is difficult to objectively define a single cause. For example, ICU patients often have multiple risk factors for diarrhea at the same time, such as low albumin, antimicrobial administration, and tube feeding, making it difficult to determine which risk is the cause of diarrhea. Therefore, physicians often evaluate and subjectively determine the suspected cause of diarrhea. Unfortunately, we have not been able to collect physician assessments on the cause of diarrhea for the patients in this study. However, the possible risks of diarrhea were listed in Table 1 (PPI, laxatives, antibiotics, antivirals, chemotherapy, enteral nutrition, etc.).

Reviewer comment

2) Were there no probiotics or synbiotics done?

Response

Thank you for this clarifying question. We do not routinely use probiotics or synbiotics against diarrhea. None of the included patients used probiotics or synbiotics. This information has been added to the Results (line 191) section, shown below:

Results (line 191) 

“None of the patients used probiotics or synbiotics.”

Reviewer comment

3) Were any drugs such as Daiken Chutou used to regulate bowel movements?

Response

We do not routinely use Daiken Chutou to regulate bowel movements. None of the included patients used this drug. 

Reviewer 2

Reviewer comment

The manuscript is quite interesting as it offers a quantitative approach to diarrhea in the ICU. However, I wonder about a few points:

1. The technique of determining the amount of stool based on a retrospective electronic medical record may not be accurate. In fact, not all ICU cases are placed on a urinary catheter, and weighing the diaper will mix feces and urine.

Response

Thank you for this insightful comment. We have additionally investigated whether a urinary catheter was inserted in the included patients. We found that 218/231 (94.4%) patients had catheters (216 patients had any forms of urine catheters and 2 patients had nephrostomy catheters), and 5/231 (2.2%) had anuria. Eight patients were voiding freely without any catheters. Because those patients could have urine contamination, we have added an analysis limited to only patients without the risk urine of contamination (patients with any catheter or anuria). As a result, the association between the quantity of diarrhea and mortality remained (adjusted RR 1.10, 95% CI 1.01 to 1.21, p=0.030). We have added the catheter information and the sensitivity analysis to the methods (lines 154-157), results (lines 191-193), Table 3, and discussion (lines 290-294).

Methods (lines 154-157)

“Fourth, to evaluate the influence of urine and stool contamination, we performed an additional analysis limited to only patients without any risk of urine contamination (anuric patients or patients with any forms of urinary catheter and/or nephrostomy).”

Results (lines 191-193)

“Some forms of urinary catheter and/or nephrostomy catheters were inserted in 218/231 (94.4%) patients (216 urinary catheters and 2 nephrostomy catheters), and 5/231 (2.2%) patients were anuric.”

Discussion (lines 290-294)

“Third, eight patients were voiding freely without the use of urine catheters. Measuring the weight of the diaper could be the sum of stool and urine in those patients; however, our sensitivity analysis showed that the increase in the risk of mortality remained, even after limiting the analysis to only patients without risk of urine contamination.” 

Reviewer comment

2. Diarrhea is the result of some medical conditions, some drugs used and even the diet. Can the authors further analyze the correlation between the amount of diarrhea and these factors?

Response

Thank you for this insightful comment. As suggested, we additionally evaluated the correlation between potential risk factors of diarrhea and the quantity of diarrhea using the linear regression model. 

The univariate linear regression analyses showed that acute kidney injury and chemotherapy were correlated with higher quantities of diarrhea. 

Beta coefficients, 95% confidence intervals, and P-values for the association between potential risk factors of diarrhea and the quantity of diarrhea.

Variable Estimate (g) 95% CI p-value

Medical condition 

Charlson comorbidity index 14.3 -15.8 to 44.4 0.35

Serum albumin 21.1 -59.4 to101.5 0.61

SOFA score -2.3 -17.1 to 12.5 0.76

Acute respiratory distress syndrome 101.2 -46.7 to 249.0 0.18

Sepsis 71.7 -42.6 to 185.9 0.22

Acute kidney injury 157.9 43.4 to 272.5 0.01

Renal replacement therapy 58.8 -73.0 to 190.7 0.38

Mechanical ventilation -9.0 -128.0 to 110.1 0.88

Noradrenaline use 41.9 -75.0 to 158.7 0.48

Drug 

Proton pump inhibitor -35.8 -195.4 to 123.8 0.66

Laxative drug -45.4 -159.8 to 69.1 0.44

Antibiotics 102.3 -116.6 to 321.2 0.36

Antivirals 47.1 -166.4 to 260.7 0.66

Chemotherapy 347.7 37.8 to 657.6 0.03

Diet 

Enteral nutrition 58.5 -64.0 to 181.0 0.35

Reviewer comment

3. Quantification of diarrhea should note the number of days of diarrhea and duration of diarrhea. In the article, only the amount of stools per day was mentioned, but not the whole process of diarrhea.

Response

We appreciate these helpful suggestions. To clarify this information, we reviewed two different durations of diarrhea. We investigated the number of consecutive days of diarrhea since the first occurrence, and the total days of diarrhea during ICU admission, including intermittent diarrhea. The median number of consecutive days of newly developed diarrhea was 1 day (IQR [1, 2]) and the median total number of days of diarrhea in the ICU was 2 days (IQR [1, 4]). We have added this information to the Result (lines 195-197) section and Table 1. 

Results (lines195-197) 

“The median consecutive days of diarrhea was 1 day (IQR [1, 2]), and the median total number of days of diarrhea in the ICU was 2 days (IQR [1, 4]).”

Additionally, we conducted analyses to evaluate the association between the duration of newly developed diarrhea and in-hospital mortality, and the total number of days of diarrhea in ICU and in-hospital mortality. The multivariable analysis showed that the point estimates were similar to the primary analysis, but there was no statistical significance.

Risk ratios, 95% confidence intervals, and P-values for the association between the duration of newly developed diarrhea and in-hospital mortality, and the total number of days of diarrhea in ICU and in-hospital mortality.

 Unadjusted Adjusted

　 RR 

[95% CI] p-value RR 

[95% CI] p-value

Consecutive days of newly developed diarrhea (per one-day increase) 1.13 

[1.03, 1.24] 0.01 1.09 

[1.00, 1.20] 0.06

Total number of days of diarrhea (per one-day increase) 1.06 

[1.00, 1.14] 0.04 1.05 

[0.98, 1.21] 0.17

Adjusted with CCI, SOFA score, and serum albumin.

Reviewer 3

Reviewer comment

Thank you for the opportunity to review this interesting manuscript. The authors used a retrospective cohort to examined the association between diarrhea quantity and mortality. Below are a few comments intended to enhance the manuscript.

1.Incidence of diarrhea- the authors report the incidence of diarrhea in the ICU 10-78%. The studies quoted are older studies. The most recent Prospective Cohort study (Dionne et al, ICM 2022) showed the incidence of diarrhea in the ICU to be between 35-70%. Would suggest updating this.

Response

Thank you for your comment. As suggested, we have changed the range of incidence of diarrhea (Introduction line 55), as below:

Introduction (line 55)

“Diarrhea is a common gastrointestinal symptom in the intensive care unit (ICU), with an incidence of 35%–70% [1].” 

Reviewer comment

2. The authors state that CDI is a common cause of diarrhea in the ICU. However, both prospective and retrospective cohort studies show a rate of 2.1%. The present cohort it was <1%. Suggest rephrasing.

Response

Thank you for pointing this out. As suggested, we have revised this sentence (Introduction lines 56-59 and Discussion lines 259-261), as below:

Introduction (lines 56-59)

“In ICU patients, enteral nutrition (composition, osmolarity, speed, intermittent or continuous, and fiber), drugs (e.g., antibiotics, laxatives), infectious diseases (e.g., Clostridium difficile infection [CDI]), and comorbidity (e.g., anemia, cirrhosis) can cause diarrhea”

Discussion (lines 259-261)

“Patients with CDI or cytomegalovirus enteritis, which are known to cause diarrhea, have been reported to have higher mortality, but they were excluded from our study.”

Reviewer comment

3. The most recent study (PMID: 35411491) did not show an association between mortality and diarrhea. Many of the studies that demonstrated this relationship where retrospective in nature. A point that should be highlighted in the current manuscript.

Response

Thank you for this suggestion. Firstly, we have added the limitation of a retrospective design in the discussion section (lines 297-300). 

Discussion (lines 297-300)

“Finally, this was a retrospective single-center study. Prospective multicenter studies that consider the diarrhea quantity are needed to improve the measurement validity of diarrhea and the generalizability of these findings.”

We agree with the reviewer's comment that we should highlight the differences from the most recent study by Dionne et al (ICM 2022;48(5):570-9). Some prospective and retrospective observational studies have shown an association between diarrhea and mortality (Arquivos de Gastroenterologia 2008;45(2):117-23, Scientific Reports. 2016;6(1):24691). Other studies, such as the study by Dionne et al, have shown no significant difference between diarrhea and mortality. Although the retrospective design may explain some of the differences in results, the results of our study suggest that the inconsistent results may be because most studies did not consider the quantity of diarrhea in their assessment. A study by Trilapuret et al. using the diarrhea definition that included quantity showed an association between diarrhea presence and mortality (Scientific Reports. 2016;6(1):24691). We also found an association between diarrhea quantity and mortality. Future prospective studies taking into account the quantity of diarrhea could help clarify this association.

As suggested, we have included this point in the discussion (lines 248-255), as below:

Discussion (lines 248-255)

“A recent prospective study evaluating the association between the presence or absence of diarrhea, as defined by the WHO definition of at least 3 liquid bowel movements per day, found that mortality was not associated with the presence of diarrhea [1]. One possible reason for the inconsistent results of these studies may be that diarrhea is defined by consistency and frequency, without taking quantity into account, which might lead to classifying small quantities of clinically unimportant bowel movements as diarrhea.” 

Reviewer comment

4. The authors describe different definitions of diarrhea throughout the manuscript-- Bristol >6, WHO definition of >3 liquid BM per day, then quote the ESICM of >200g. Suggest clarifying this and harmonizing it in the manuscript

Response

Thank you for pointing out this inconsistency. We appreciate the opportunity to clarify this point. We defined diarrhea by the WHO definition (three or more loose or liquid stools per day). According to the WHO definition, consistency (loose or liquid) must be evaluated, but the exact methods for assessing consistency are not clearly defined. Therefore, the Bristol Stool Chart Scale, which has been well-validated, was used to evaluate the consistency. In addition, to make the analysis of the association between diarrhea quantity and outcome easier to understand, the results are presented as a risk ratio for a per 200 g increase in diarrhea quantity. The reason for the 200 g intervals is that the ESICM uses a 200 g cutoff for diarrhea quantity. 

We have revised the manuscript to clarify these points (Abstract lines 35-38, Methods lines 113-114, 116-117, 120-121, and line 140). 

Abstract (lines 35-38)

“Newly developed diarrhea was defined according to the World Health Organization definition. The consistency of diarrhea was evaluated by the Bristol stool chart scale, and the quantity of diarrhea was assessed on the day when patients newly developed diarrhea.”

Methods (lines 113-114)

“We defined diarrhea by the WHO definition (three or more loose or liquid stools per day).”

Methods (lines 116-117)

“The Bristol Stool Chart Scale (BSCS) was used to evaluate the consistency of each stool sample [27].”

Methods (lines 120-121)

“A BSCS of 6 or 7 is classified as diarrhea [28, 29]” was removed.

Methods (line 140)

“ESICM uses a 200–250 g cutoff for diarrhea quantity.”

Reviewer comment

5. The authors highlight the limitations of the study, however, suggest expanding the discussion to include areas for further research to help answer the remaining questions in this area (ex. mechanism for diarrhea in the ICU) and how this research may be done.

Response

We appreciate these helpful suggestions. We have updated the discussion section to include areas for further research (lines 294-300). 

Discussion (lines 294-300)

“Fourth, mechanistic data on developing diarrhea in ICUs were lacking. To elucidate mechanisms, studies are needed to assess the relationship between changes in the microbiome over time during ICU stays with newly developed diarrhea, and to evaluate whether interventions to correct the microbiome improve diarrhea. Finally, this was a retrospective single-center study. Prospective multicenter studies that consider diarrhea quantity are needed to improve the measurement validity of diarrhea and the generalizability of these findings.”

Reviewer comment

6. The Mortality for diarrhea in this study was 20%. It would be helpful to the reader to compare that to typical ICU stay or compare it to the patients in the study who did not develop diarrhea.

Response

Thank you for this helpful suggestion. ICU mortality, in-hospital mortality, ICU length of stay, and hospital length of stay for patients without diarrhea (patients that did not meet the inclusion criteria), and those with diarrhea are shown in the table below: 

S1 Table. The mortality and length of stay of the entire cohort

 Total 

n=1579 Non-diarrhea† 

n=1245 Diarrhea‡ 

n=334

ICU mortality, n (%) 103 (6.5) 69 (5.5) 34 (10)

In-hospital mortality, n (%) 212 (13) 126 (10) 86 (26)

ICU LOS, median [IQR], day 3.0 [2.0, 5.0] 2.0 [2.0, 3.8] 7.0 [4.0, 11.0]

Hospital LOS, median [IQR], day 17 [9, 39] 14 [8, 29] 39 [19, 68]

†Patients who did not meet inclusion criteria, ‡Patients who met the inclusion criteria (including patients who met exclusion criteria). There was no missing outcome data. IQR: Interquartile range, LOS: Length of stay.

Patients without diarrhea had low ICU and in-hospital mortality, and short ICU and hospital length of stay.

We have added this information to the results (lines 171-172) and the S1 Table.

Results (lines 171-172)

“During the study period, 1579 patients were admitted to the ICU, and 334 adult patients with newly developed diarrhea were included in this study (S1 Table).”

We appreciate all the comments, which in our opinion have greatly helped us improve the paper.

---

## [Decision Letter · Decision Letter 1]

10 Jan 2023

Dose-response relationship between diarrhea quantity and mortality in critical care patients: A retrospective cohort study

PONE-D-22-18062R1

Dear Dr. Yamamoto,

We’re pleased to inform you that your manuscript has been judged scientifically suitable for publication and will be formally accepted for publication once it meets all outstanding technical requirements.

Kind regards,

Chinh Quoc Luong, MD., PhD.

Academic Editor

PLOS ONE

Additional Editor Comments (optional):

Reviewers' comments:

Reviewer's Responses to Questions

**Comments to the Author**

1. If the authors have adequately addressed your comments raised in a previous round of review and you feel that this manuscript is now acceptable for publication, you may indicate that here to bypass the “Comments to the Author” section, enter your conflict of interest statement in the “Confidential to Editor” section, and submit your "Accept" recommendation.

Reviewer #1: All comments have been addressed

Reviewer #2: All comments have been addressed

2. Is the manuscript technically sound, and do the data support the conclusions?

Reviewer #1: Yes

Reviewer #2: Yes

3. Has the statistical analysis been performed appropriately and rigorously? 

Reviewer #1: Yes

Reviewer #2: Yes

4. Have the authors made all data underlying the findings in their manuscript fully available?

Reviewer #1: Yes

Reviewer #2: Yes

5. Is the manuscript presented in an intelligible fashion and written in standard English?

Reviewer #1: Yes

Reviewer #2: Yes

6. Review Comments to the Author

Reviewer #1: Thank you for the opportunity to review the manuscript again. I read the manuscript with great interest. My concerns have been addressed.

Reviewer #2: After editing, the manuscript was more reasonable. The efforts of the authors in pursuing this topic deserve one vote in favor of the article being accepted for publication.

7. PLOS authors have the option to publish the peer review history of their article (what does this mean?). If published, this will include your full peer review and any attached files.

Reviewer #1: No

Reviewer #2: No

---

## [Editor Report · Acceptance letter]

13 Jan 2023

PONE-D-22-18062R1 

Dose-response relationship between diarrhea quantity and mortality in critical care patients: A retrospective cohort study 

Dear Dr. Yamamoto:

I'm pleased to inform you that your manuscript has been deemed suitable for publication in PLOS ONE. Congratulations! Your manuscript is now with our production department. 

Kind regards, 

on behalf of

Dr. Chinh Quoc Luong 

Academic Editor

PLOS ONE